# The stochastic logistic model with correlated carrying capacities reproduces beta-diversity metrics of microbial communities

**Silvia Zaoli** , **Jacopo Grilli** *

Quantitative Life Sciences section, The Abdus Salam International Centre for Theoretical Physics (ICTP), Trieste, Italy

* jgrilli@ictp.it

**Data Availability Statement:** This paper does not use original data. All data needed to evaluate the conclusions in the paper are available from the original references cited in the paper. The Python

## Abstract

The large taxonomic variability of microbial community composition is a consequence of the combination of environmental variability, mediated through ecological interactions, and stochasticity. Most of the analysis aiming to infer the biological factors determining this difference in community structure start by quantifying how much communities are similar in their composition, trough beta-diversity metrics. The central role that these metrics play in microbial ecology does not parallel with a quantitative understanding of their relationships and statistical properties. In particular, we lack a framework that reproduces the empirical statistical properties of beta-diversity metrics. Here we take a macroecological approach and introduce a model to reproduce the statistical properties of community similarity. The model is based on the statistical properties of individual communities and on a single tunable parameter, the correlation of species' carrying capacities across communities, which sets the difference of two communities. The model reproduces quantitatively the empirical values of several commonly-used beta-diversity metrics, as well as the relationships between them. In particular, this modeling framework naturally reproduces the negative correlation between overlap and dissimilarity, which has been observed in both empirical and experimental communities and previously related to the existence of universal features of community dynamics. In this framework, such correlation naturally emerges due to the effect of random sampling.

## Author summary

Several biological and ecological forces shape the composition of microbial communities. But also contingency and stochasticity play an important role. Comparing communities, identifying which conditions are similar in communities with similar species composition, allows to identify which forces shape their structure. Here we introduce a modeling framework which reproduces the statistical features of community similarity. We identify a single relevant parameter that captures in a single number the multidimensional nature of similarity in community composition. These results set the basis for a quantitative, and predicting, theory of the statistical properties of microbial communities.

code used to perform the analysis is available at
https://github.com/SilviaZaoli/sample_similarity.

**Funding:** The author(s) received no specific
funding for this work.

**Competing interests:** The authors have declared
that no competing interests exist.

## Introduction

A surprising large number of microbial species is found in a spoon of soil or a drop of water
sampled at a single location and time [1]. The large values of alpha-diversity parallel with high
beta-diversity: the taxonomic composition would be different if the sample were collected at a
different time or in a different location [2].

A primary objective of microbial ecology is to link the observed variability of taxonomic
composition with its mechanistic causes. In order to achieve this goal, since the first environ-
mental assays of the late 80s to today's large sequencing efforts, one of the main goals of micro-
bial communities data-analysis has been to disentangle the predictable, replicable variation of
community composition—the "signal"—to contingent, non-replicable, uninformative, vari-
ability—the "noise". Methods to identify replicable temporal or spatial patterns in the change
of community composition typically rely on some measure of dissimilarity between communi-
ties—or equivalently, on a beta-diversity metric. Such a measure allows in fact to define a "dis-
tance" metric between communities, which can be ultimately used to compare and cluster
communities. For example, a commonly used model-free approach is Principal Coordinate
Analysis [3], which takes as input a matrix of sample-to-sample distances and identifies the
coordinates which explain most of the variation between samples. This method allows to infer
which variables are a more relevant source of variation and to identify clusters of similar sam-
ples. For instance, at the global scales, clusters identified comparing samples of microbial
communities from different environments all around the world are well explained by the envi-
ronment type [1]. At a smaller scale, the composition of gut microbial communities is associ-
ated with host clinical markers and lifestyle factors [4].

Despite the centrality of similarity and beta-diversity metric in microbial ecology analysis
pipelines, we lack a mechanistic understanding of which aspects of community variability
influence their values. Here we aim at formulating a quantitative phenomenological frame-
work able to reproduce the observed statistical properties of community similarity. The dis-
similarity between two communities is in fact caused both by signal and by noise, but we miss
a modeling framework that can be used to assess the effect of each. The sampling nature of the
data also has a strong effect on several beta-diversity metrics [5], as it adds an additional source
of noise and a bias in the observations, and should be explicitly considered.

Macroecology is a promising avenue for filling this gap. By characterising the statistical
properties of community composition, macroecology provides access to quantities that are
reproducible across systems. In perspective, a macroecological approach could allow to disen-
tangle the statistical property of contingent, non-reproducible, noise from the reproducible
statistical features of environmental variability.

Most of the efforts in macroecology have been focused on describing and predicting alpha-
diversity. For instance, the species abundance distribution (SAD) of empirical microbiome is
well characterized across ecosystems [6]. The abundance fluctuation distribution (AFD) is well
described by a Gamma distribution, while the distribution of the mean abundance of species is
typically well described by a Lognormal [7]. One can extend the macroecological description
to dynamics, and characterise the variability of species abundance and diversity across time-
scales [8, 9].

One of the examples of the study of beta-diversity under a macroecological perspective is
given by the dissimilarity-overlap analysis (DOA) [10], where beta-diversity metrics have been
used to infer ecological mechanisms underlying the differences in composition between sam-
ples. The DOA is based on two beta-diversity measures, dissimilarity and overlap. The dissimi-
larity between two communities measures the differences of the relative abundances of the
species present in both samples. The overlap measures the probability that, if we pick an

individual from one of the two samples, it belongs to a species that is present in both samples. These two metrics capture, in principle, two distinct aspects of community variability and should therefore vary independently. However, when they are plotted one against the other for a set of samples—in what is termed the Dissimilarity-Overlap curve (DOC)– both natural [10] and experimental [11] communities display a decreasing pattern: communities with high overlap tend to have low dissimilarity. The robustness of this DOC pattern suggests that it could be explained by a robust, general, process. The leading interpretation is that a decreasing DOC is a consequence of the universality of the dynamics [10]: different communities are subject to the same ecological dynamics, characterised by the same parameters, and differ because they occupy different dynamical attractors. Other studies have shown that other mechanism than universal dynamics might be responsible of a decreasing DOC [12]. One limit of these observations and their interpretation is that they are mostly qualitative. For instance, both the empirical DOC and models based on environmental gradients [12] produce negative DOCs. But are the empirical and modeled DOCs in quantitative agreement? It is not trivial to answer this question, because most of the available (null) models cannot be easily parameterized using the available data.

More generally, one could generalize the DOA by comparing the values of the several existing beta-diversity metrics. Empirical values of different beta-diversity measures are in fact correlated [13], but, again, we lack a quantitative understanding of their relationship.

Here, we introduce a model of community composition that is able to reproduce quantitatively the empirical values of beta-diversity and the relationship between different beta diversity metrics. The model includes two sources of variability for microbial communities. First, temporal stochasticity, corresponding to the non-reproducible sources of variation. This variability was shown to be well described by the stochastic logistic model (SLM), a model that describes the temporal evolution of species abundances under a stochastic environmental noise [7, 14]. According to this model, species abundances fluctuate in time around a constant typical abundance. The second source of variability concerns how this typical abundance differs across communities [9], which represents the reproducible part of variability. The difference between these two sources, which in our model is controlled by varying a single parameter, can be of a larger or lower magnitude. Both sources of variability are modelled phenomenologically, and several mechanisms could underlie them. For instance, ecological interactions contribute to both, as they may mediate rapid abundance fluctuations and be the origin of alternative stable states in community dynamics. Environmental factors also contribute to both, either in the form of rapid environmental fluctuation or of differences in the overall environmental conditions across communities. Importantly, our model also incorporates explicitly the sampling process, allowing us to study the effect of sampling on the relationship between beta-diversity metrics and, in particular, on the DOC.

We compare the model predictions with empirical data of the human gut microbiome of different human hosts (see Methods). The model, by varying the parameter measuring the difference across-communities, jointly reproduces several beta-diversity metrics both within and across hosts. As a consequence, it also reproduces quantitatively Dissimilarity-Overlap curves, uncovering how random sampling introduces a relationship between these two—in principle, but not in practice—independent metrics.

## Results

### A model for community composition with tunable similarities

The model for community composition that we propose is based on the statistical properties of empirical microbial communities which describe how they change across time and space.

The stationary fluctuation of an OTU $i$ abundance $\lambda_i$ in time are well described by the stochastic logistic model (see [7] and Methods). Consequently, at stationarity the abundance is Gamma distributed

$$P(\lambda_i; K_i, \sigma_i) = \frac{1}{\Gamma(2/\sigma_i - 1)} \left(\frac{2}{\sigma_i K_i}\right)^{\frac{2}{\sigma_i}-1} \lambda_i^{\frac{2}{\sigma_i}-2} e^{-\frac{2}{\sigma_i K_i}\lambda_i}, \tag{1}$$

where $K_i$ is directly related to the carrying capacity of an OTU $i$, and $\sigma_i \in [0, 2)$ is related to the level of environmental variability (see Methods). Given the compositional nature of the data, the carrying capacity of OTUs cannot be estimated from the data, as the total abundance is not known. It is however possible to show that the values $K_i$ are proportional to the carrying capacity, up to an unknown proportionality constant, which is sample-specific and common to all species in that sample [7, 9]. While the proportionality constant is unknown, one can effectively ignore its value as it does not impact the properties of the fluctuations of species abundance [7]. For simplicity, with this important caveat in mind, we will still refer to $K_i$ as carrying capacity in the following. The values of $K$ and $\sigma$ for each OTU can be estimated from the time series of its abundance (see Methods). Our previous analyses [9] showed that the value of $K$ for an OTU remains constant for long stretches of time. Over timescales of weeks, Eq 1, together with a set of values of $K$ and $\sigma$ characterises the time-variability of composition.

Comparing the composition across hosts, it is known that differences in $K$, together with random fluctuations of abundance, explain almost all the dissimilarity between hosts [9]. Differences in the values of $\sigma$ of each OTUs between communities, instead, have a much smaller role in differentiating communities. Values of $K$ estimated from the time series of two different hosts are correlated at various degrees (see Tables B, C, D, and E in S1 Text), but always at a significant level. This shows that the carrying capacity is to some extent characteristic of the OTU. As expected however [9] the correlation is lower than the one obtained by comparing two segments of the time-series of a given individual. This indicates that different communities, and the same individual over time, have values of $K$ that are different, although highly correlated.

The distribution of $K$ is lognormal above a threshold (Fig 1A, dashed lines are the threshold) [7]. The existence of a threshold is due to sampling. In fact, if $N_s$ is the sampling depth, OTUs with values of $K$ close to or below $1/N_s$ might not be sampled. While this is a probabilistic effect, we approximate it as an hard cutoff [7]. We fit therefore a truncated lognormal distribution (black line in Fig 1A) to all the $K > c$, with $c$ a threshold different for the different environments. For all environments, the fitted lognormal describes well the data above the threshold. It is important to notice that, since the values $K_i$ vary over-time, the inferred distribution capture both the intrinsic variability across OTUs and the variability of each OTU over time. As shown in [9], both these sources of variability are lognormally distributed and combine in a multiplicative way, resulting in a overall lognormal distribution for the combined effect.

We also explored the heterogeneity of environmental variability $\sigma$ across species, which has been previously neglected. We find that the distribution of $\sigma^2$ is exponential with mean 0.93 (see Fig 1B).

To formulate our model, we start by observing that different communities are characterized by the same parameters of the lognormal distribution of carrying capacities and of the exponential distributions of variability $\sigma^2$. See, for example, the distributions corresponding to the gut communities of different individuals in Fig 1A and 1B. Therefore, we assume that $K$ and $\sigma$ are identically distributed across communities. We further assume that the total number $S$ of

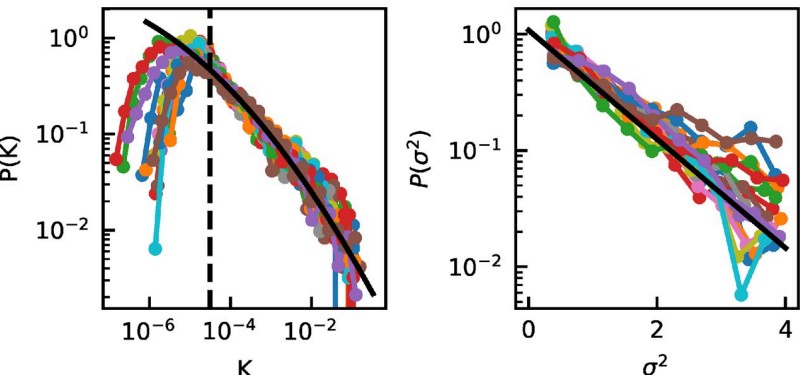

**Fig 1. Parametrization of the model.** A) Distributions of $K$ for each individual. Colored points and lines are normalized histogram of the data. The black line is a maximum likelihood fit to a truncated lognormal distribution, with truncation at $10^{-4.5}$, of the values from all individuals (parameters of fitted lognormal: $\mu = -19.85$, $s = 4.93$). The dashed line refers to the cutoff due to finite sampling: below this value the effect of sampling produces deviation from the Lognormal distribution; B) Distribution of $\sigma^2$ for each individual (colored points and lines are normalized histogram of the data). The black line is a maximum likelihood fit to an exponential distribution of the values from all individuals (mean = 0.93). All individuals are characterized by the same distributions of carrying capacity $K$ and variability $\sigma$.

OTUs present in a community, including the ones that are present but undetected, is the same across communities.

Then, the aim of our model is to generate pairs of communities with different levels of similarity. Based on the previous observations on the variability of $K$ and $\sigma$ across communities [9], we assume that an OTU $i$ in two different communities have the same values of $\sigma_i$ but different values of $K_i$. The correlation between the logarithms of the carrying capacities, $\rho_K$, obtained by averaging across OTUs, tunes the level of similarity of two communities.

To generate a pair of communities, we generate $S$ pairs of values from a bivariate gaussian distribution with mean and variance corresponding to the ones of $\log K$ and with correlation $\rho_K$. Each pair of values corresponds to an OTU $i$, and represents the logarithms of its carrying capacity $\log K_i$ in the two communities. For each OTU $i$ we also extract a value of $\sigma^2$, common to the two communities, from the corresponding exponential distribution. Given the set of $K$ and $\sigma$ values for a community, we extract the real abundance $\lambda_i$ of OTU $i$ from the Gamma distribution in Eq (1) with parameters $K_i$ and $\sigma_i$. We finally extract a finite sample of the community with total number of reads $N_{reads}$ from a multinomial distribution with $N_{reads}$ trials.

## By tuning the correlation of carrying capacity the model predicts a wide range of values of beta-diversity

We use the model formulated in the previous section to generate in-silico communities with a range of values of community similarities, obtained by varying the correlation between carrying capacities $\rho_K$. We use this ensemble of communities to study the value of different beta-diversity measures.

In particular, we simulate pairs of samples from different communities and from the same community at different times, by tuning the value of $\rho_K$. For each pair we compute six different beta-diversity measures: i) Jaccard similarity, ii) Sørensen similarity index, iii) Whittaker index, iv) Morisita-Horn similarity, v) Bray-Curtis dissimilarity, and vi) Horn similarity. The choice of these commonly used different measures is aimed at covering different types of measures, accounting for presence-absence, abundance, or both, and more or less focused on common species rather than rare ones. These different beta-diversity measures have correlated

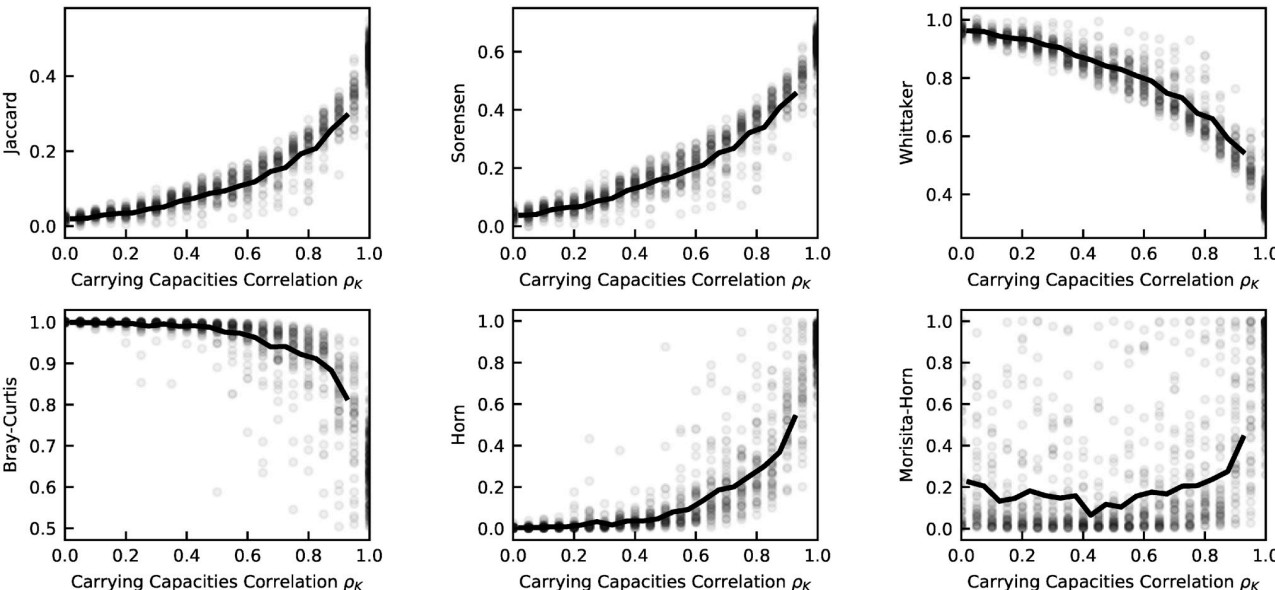

**Fig 2. Relationships between dissimilarity measures for communities generated with the model.** The different dissimilarity measures (A: Jaccard similarity, B: Sørensen index, C: Whittaker index D: Bray-Curtis disimilarity, E: Horn similarty, F: Morisita-Horn similarity) are plotted against the Carrying capacity correlation $\rho_K$. Grey circles represent the 200 pairs of communities generated with the model. Each community has $S = 10^4$ OTUs. Each pair of communities have the same $\sigma$, extracted from an exponential distribution with mean 0.9. Values of $K$ are extracted from a lognormal distribution with parameters $\mu = -19$ and $s = 5$. 100 pairs have the same values of $K$, to mimic samples from the same community at different times. The remaining 100 pairs have correlated values of $K$, with $\rho_k$ ranging between 0.5 and 1, obtained by exponentiating values extracted from a bivariate Gaussian distribution. For each community, abundances are extracted from a Gamma distribution with parameters $K$ and $\sigma$. For the pairs with the same values of $K$, Gamma-distributed abundances have a correlation ranging from 0 to 0.5. Reads are obtained from the real abundances by simulating multinomial sampling with number of reads $3 * 10^4$. Black lines are the binned average of the grey circles.

values in the synthetic communities (see Fig 2, where the measures are all plotted against the input value of $\rho_K$). The correlation can be positive or negative because some metrics measure similarity and other dissimilarity. Note that the maximal level of correlation $\rho_K = 1$ does not correspond to the theoretical maximum level of similarity (e.g. 1 for Jaccard or 0 for Bray-Curtis). Each OTU is in fact also subject to the stochastic environmental fluctuations of the Stochastic Logistic model, which are by definition independent across communities. This level of fluctuation, together with the effect of finite sampling, contribute to a decreased level of similarity.

## The model reproduces the empirical relationships between dissimilarity measures

We compare the empirical relationships between beta-diversity measures with the ones obtained from the model. A critical problem in doing the comparison is that the values of $K$ are not known for individual samples, and therefore, the value of $\rho_K$ is unknown for any pair of samples.

In order to circumvent this problem we used our model to infer the value of $\rho_K$ in any pair of samples (see Methods). We notice that, in our in-silico data, there is a simple relationship between the Spearman correlation of abundances of a pair of sample and their value of $\rho_K$ (see S1 Fig). By characterizing this relationship, we are able to infer the value of $\rho_K$ of a pair of samples.

Given the inferred value of $\rho_K$ we then generate a pair of in-silico samples. In addition to $\rho_K$, to fully specify the in-silico data, we used the fitted distributions of $K$ and $\sigma$, a number of

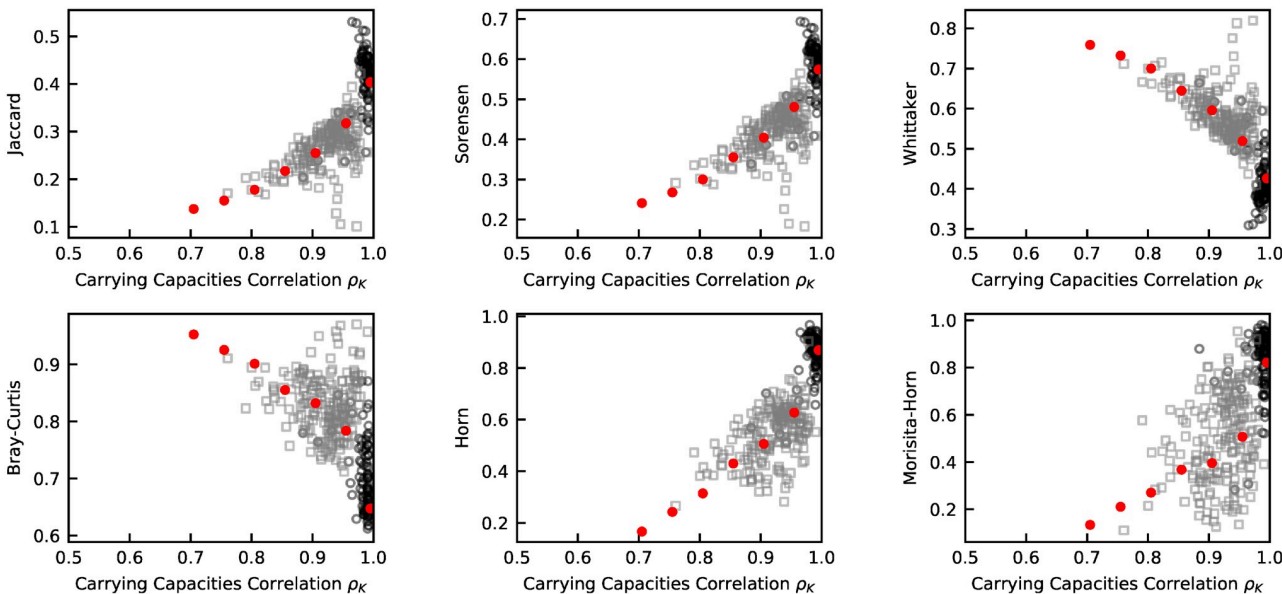

**Fig 3. Comparison between the relationships between dissimilarity measures in empirical data and according to the model.** The different dissimilarity measures (A: Jaccard similarity, B: Sørensen index, C: Whittaker index D: Bray-Curtis dissimilarity, E: Horn similarty, F: Morisita-Horn similarity) are plotted against the carrying capacity correlation $\rho_K$. Black circles correspond to pairs of empirical samples from the same host, while grey squares correspond to pairs of empirical samples from different hosts (but of the same dataset). The value of $\rho_K$ for individual samples is inferred as specified in the Methods. Red dots are the binned average of the predictions of the model. The model is simulated with the distributions of $K$ and $\sigma$ fitted from the data, and with a number of species equal to the one estimated (see Section B in S1 Text). The number of reads is equal to the average number of reads for the empirical samples, $3 \cdot 10^4$.

reads equal to the average of the reads in the empirical samples, and a number of OTUs $S$ equal to the number estimated (see SI section 2). The relationships predicted by the model follow quantitatively the empirical patterns (Fig 3).

This suggests that the ingredients of the model are sufficient to capture the statistical features of communities and of differences between communities that are relevant to determine the relationships between beta-diversity measures. In Table A in S1 Text, we report the proportion of variation observed in the data for the different beta-diversity metrics explained by our model.

## Overlap-dissimilarity negative relationship is expected under finite sampling

In this section we consider the two beta-diversity metrics, Dissimilarity and Overlap, introduced in [10] (see Methods). Our model shows that a decreasing Overlap-Dissimilarity curve can emerge purely because of sampling, and therefore might not reflect any ecological property of the communities.

For the communities generated with the model, prior to sampling, overlap and dissimilarity are completely independent. In fact, all pairs of communities have overlap equal to 1, as all species are always present. However, after simulating the sampling, a non-trivial Dissimilarity-Overlap curve emerges (Fig 4A). For pairs of communities with a medium to high correlation $\rho_k$ between their (log) values of $K$, the Dissimilarity-Overlap curve has a decreasing pattern. The two insets of panel A clarify why this pattern emerges. The insets show the abundances of two pairs of communities, one with a high $\rho_k$ and one with a low $\rho_k$. The dissimilarity of the pair of samples is computed on the OTUs that are sampled in both communities, i.e. the blue

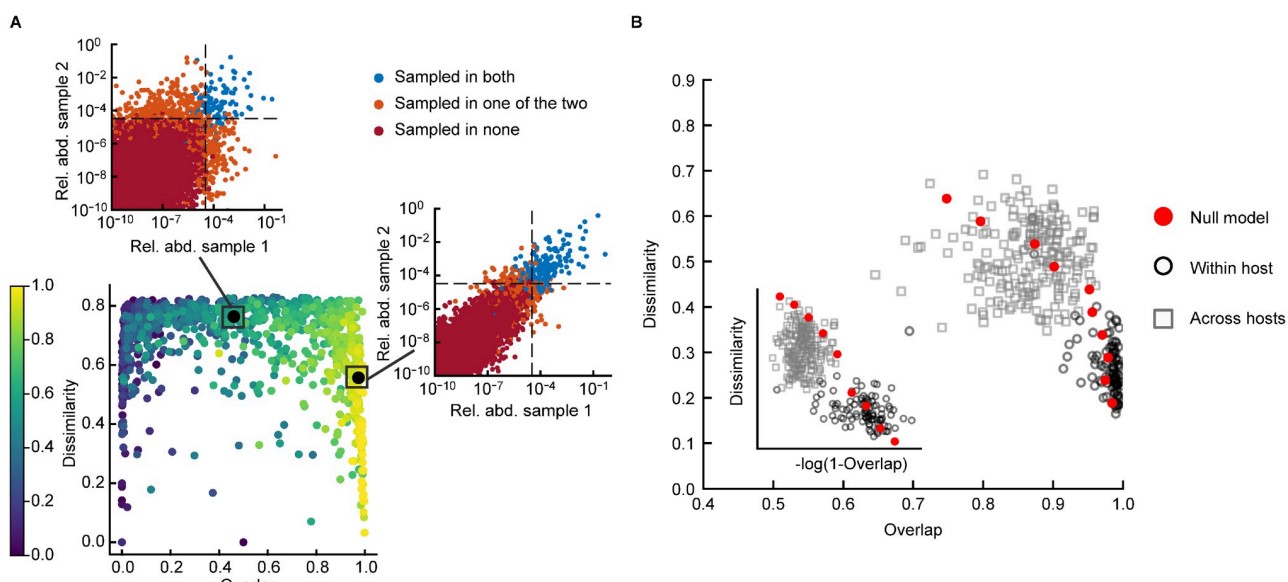

**Fig 4. Overlap-Dissimilarity curves in the model and in empirical data.** A) Relationship between Overlap and Dissimilarity for communities generated with the model. The color of the circles corresponds to the correlation $\rho_k$ of the values of $K$ of the community pair, ranging from 0.6 to 1. The two insets show scatter plots of the abundances in two pairs of communities, one with a high $\rho_k$ and one with a low $\rho_k$. Blue circles represent OTUs sampled in both communities, orange circles OTUs sampled in only one community, and red circles OTUs sampled in neither. The dotted lines mark $1/N_{reads}$. B) Relationship between overlap and dissimilarity in empirical data (black circles: samples from the same hosts, grey squares: samples from different hosts) and according to the model (red circles, binned average of model prediction). The inset shows the same plot with a logarithmic scale on the x axis. For the main plot, the binned average of the model prediction is performed along the y axis, to better capture the pattern at high overlap values.

points. It is clear from the insets that the abundances of the blue OTUs are more dissimilar when $\rho_k$ is low. The overlap, instead, depends on the abundance of the OTUs observed in both samples (blue) with respect to that of all the observed ones (blue + orange). This quantity also has a clear pattern with $\rho_k$, because the proportion of orange OTUs diminishes when $\rho_k$ increases. These two effects, caused by sampling, create the the decreasing pattern in the Dissimilarity-Overlap curve.

To verify if the model can quantitatively explain the Dissimilarity-Overlap curve of empirical data, we compare the empirical pattern with that obtained from the simulation of the model. The model is able to reproduce well the Dissimilarity-Overlap curve of empirical data (Fig 4B). This result indicates that the relationship created by sampling between dissimilarity and overlap is sufficient to explain the decreasing pattern seen in empirical data.

## Discussion

In this paper, we have introduced a modeling framework to describe the variability of community composition in space and time. The model is based on the stochastic logistic model (SLM), which describes the time variability and identifies the parameters that characterize a community, that is, the carrying capacity and the noise intensity. Our framework allows to model pairs of community with different levels of similarity by setting the correlation of their carrying capacities. The model quantitatively reproduces the values of several beta-diversity metrics, which weight in different ways the statistical properties of community composition variability. In particular the model naturally reproduces the negative relationship between overlap and dissimilarity observed in empirical [10] and experimental [11] data.

Our framework is based on several assumptions. In particular, it assumes that the SLM describes well the dynamics and properties of communities and that the difference in the abundance of species across communities can be captured by considering two SLMs with different carrying capacities. These two assumptions have been extensively studied in previous works. The SLM reproduces the dynamical properties of microbial communities as well as several macroecological patterns observed in empirical data [7, 14]. The variation of carrying capacities suffice to explain the typical difference of composition between communities [9].

A novel assumption—and result—of this work is that the differences in similarity observed across pairs of communities can be fully described and collapsed in a single parameter: the correlation between the (log) carrying capacities in the two communities, $\rho_k$. The fact that, by varying this parameter, one reproduces the relationship between different beta-diversity metrics is a direct test of this assumption. In principle, other properties could matter to differentiate communities. For instance, the amplitude of abundance fluctuations $\sigma$ could in principle differ across communities and be important to explain the observed beta-diversity. Our analysis complements the results of [9] by showing that the variability in $\sigma$ is negligible from a macroecological perspective. Another element that could in principle determine beta-diversity is the set of species present in a community, which could differ across community. However, our model shows that differences in carrying capacity, together with finite sampling, are sufficient to explain empirical beta-diversity values without the need to assume that different communities have different presence-absence patterns.

An interesting aspect of the variability $\sigma$, that we unveil in this paper, is that its values have a reproducible distribution across species. Interestingly, $\sigma^2$ is exponentially distributed, with a similar scale parameter across individuals. This novel macroecological patterns adds to the list of reproducible statistical properties of microbial communities, that a comprehensive theory should be able to reproduce.

Our model does not address the biological origin of the values of the parameters and of the correlation of carrying capacities across individual. Their variation is due, potentially, to multiple biotic and abiotic factors, and to the interactions between species. Our framework clarifies that they are the effective dynamics parameters that suffice to explain the statistical properties of community composition.

Once the macroecological patterns are taken into account and the SLM is used to generate in-silico communities, one naturally reproduces the empirical values of beta-diversity metrics. While our model has proven flexible (can reproduce a wide range of values of similarity) and accurate (reproduces quantitatively the empirical values), we have not studied here the fluctuations of the beta-diversity metrics and whether their joint distribution is captured by our model. Most likely, the several mechanism that the Stochastic Logistic model neglects contribute to the shape of these distributions. Characterizing these deviations from our model and linking them to the ecological forces that we did not consider in this work is the natural step forward.

Of particular relevance is the ability of our model to reproduce the empirical Dissimilarity-Overlap curve (DOC), which raises questions on its interpretation. Bashan et al. [10] interpret the empirical Dissimilarity-Overlap curve as the consequence of the fact that species dynamics is governed by the same equations and the same parameters across communities. In this view, two communities with a similar set of species (high overlap) would have similar stable states (low dissimilarity) and vice versa, producing a DOC with negative slope. Our analysis points to an alternative origin of the empirical DOCs. For communities described by non-universal parameters (correlated but different carrying capacities), a DOC with negative slope naturally emerges due to finite sampling. Consequently, the DOC would have no implication on underlying ecological mechanisms.

Beyond the interpretation of the DOCs, our results speaks directly to the original assumptions of the dissimilatity-overlap analysis. The leading assumption behind it is that the two measures of dissimilarity and overlap are independent. This statement about independence is always defined only in the light of a model for community composition: given a null statistical ensemble, the value of the dissimilarity is not correlated to the one of overlap. The model considered in [10] implicitly assumes that the typical abundance of a species and its occupancy (how likely the species is to be present) are independent, which is not verified in the data. Occupancy and average abundance do in fact display a strong correlation [15], which is predicted by the SLM with finite sampling [7]. By including explicitly sampling noise, the SLM reproduces the relation between occupancy and abundance observed in the data [7] and qualifies therefore as more reasonable model to test the independence between overlap and dissimilarity. Once the effect of sampling is included, and therefore the non-independence between abundance and occupancy is taken into account, a negative correlation between dissimilarity and overlap naturally emerges (as shown in Fig 4).

One additional interesting prediction of our framework, is that, when samples with small enough overlap are included, the DOC curve should take the shape of an inverse U (Fig 4). Therefore, for small enough values of the overlap, our model predicts positive DOCs. In empirical data, the range of values of overlap is not large enough to observe this trend. However, in experimental data [11], by considering in vitro communities grown in different nutrients it is possible to reach smaller values of overlap. In those cases an inverse-U DOC is in fact observed, as predicted by our framework.

Our results add an important step to the quantitative understanding of the structure of microbial communities. By generating more and more realistic in-silico communities, we gain an understating of what salient features of the data are the direct results of relevant biological and ecological processes, therefore allowing to disentangle general processes from contingent factors, signal from noise.

## Methods

### Data

We analyze gut microbial communities. We consider time-series of 14 individuals coming from three different datasets: ten individuals of the BIO-ML dataset [16] (all those for which a dense long-term time-series is available), the two individuals M3 and F4 from the Moving Pictures dataset [17] and the two individuals A and B from [18]. The length of the time-series ranges from 6 months to 1.5 year, and the sampling frequency varies (daily in the most dense series). Individuals A and B from [18] both undergo a period of strong disturbance to their gut flora due, respectively, to two diarrhoea episodes during a travel abroad and a *Salmonella* infection. We exclude these periods from the analysis and consider for each individuals two separate time-series, before and after the perturbation. Only samples with a number of reads $N_{reads} > 10^4$ are used. Detail on how the raw data were analyzed can be found in the Supplementary Information of [9].

### Statistical properties of the fluctuations of abundances

The stationary fluctuations of OTUs abundance have been shown to follow a Gamma distribution [7]. Additionally, dynamical properties of these fluctuation suggest that abundance dynamic can be described by a Stochastic Logistic Model with environmental noise [7, 14]:

$$\dot{\lambda} = \frac{1}{\tau}\lambda\left(1 - \frac{\lambda}{K}\right) + \lambda\sqrt{\frac{\sigma}{\tau}}\xi(t), \tag{2}$$

where $\xi(t)$ is Gaussian white noise. This model has three parameters: $\tau$ has the dimension of a time, and determines the time-scale of relaxation to stationarity, $K$ would be the carrying capacity in the absence of noise, and $\sigma$ measures the intensity of the environmental noise. If $\sigma < 2$, the model has a stationary distribution which is the Gamma distribution in Eq (1). The mean of the stationary distribution is $\langle \lambda \rangle = K\frac{2-\sigma}{2}$ and the variance $var(\lambda) = \frac{\sigma \langle \lambda \rangle^2}{2-\sigma}$. The coefficient of variation $\sqrt{var(\lambda)/\langle \lambda \rangle^2}$ depends only on the parameter $\sigma$, which can thus be interpreted as the amplitude of the fluctuations. Grilli (2020) showed that Taylor's Law applies to abundance fluctuations with exponent 2, that is, $var(\lambda) = C \cdot \langle \lambda \rangle^2$ with $C$ a constant, which implies that $\sigma$ is not correlated with $K$.

## Estimation of $K$ and $\sigma$

The parameters $K$ and $\sigma$ can be estimated from the mean and variance of the abundance time-series, inverting the expressions for the mean and variance of the stationary distribution. To estimate the variance of abundance from the sampled abundance, we need to use an expression corrected for the sampling bias. In fact, the variance of the sampled abundance is a result of the actual variance of abundance and of the variance due to the random sampling. We use the sampling-corrected estimate as done in [7]

$$var(\lambda) = \frac{1}{|T|} \sum_{t \in T} \frac{x_i(t)(x_i(t)-1)}{N(t)(N(t)-1)} - \left( \frac{1}{|T|} \sum_{t \in T} \frac{x_i(t)}{N(t)} \right)^2. \qquad (3)$$

We note that the variance estimated with this formula may result negative if many counts are 0 or 1. The OTUs for which this happens are excluded from the analysis, as it is not possible to estimate their parameters.

## Estimate of $\rho_K$ between two samples

We estimate the value of $\rho_K$ by calculating the Spearman correlation of abundances between two samples. In our in-silico data, we found that the carrying capacity correlation $\rho_K$ is in an approximate quadratic relationship with the Spearman correlation coefficient (see S1 Fig). By using this relationship found in the in-silico dataset, we can infer the empirical value $\rho_K$ of a pair of data samples.

## Beta-diversity measures

We consider six beta-diversity measures pairwise beta-diversity measures: i) Jaccard similarity, ii) Sørensen index, iii) Whittaker index, iv) Bray-Curtis disimilarity, v) Horn similarty, vi) Morisita-Horn similarity. We also included the Dissimilarity and Overlap introduced in [10]. The choice of these eight different measures is aimed at covering different types of measures, accounting for presence-absence, abundance, or both, and more or less focused on common species rather than rare ones.

   Let $n^A$ and $n^B$ be the OTU counts in two samples $A$ and $B$, with total number of reads, respectively, $N^A$ and $N^B$. Let $x = n/N$ be the sampled relative abundances. Let $S^A$ and $S^B$ be the sets of OTUs observed, respectively, in sample $A$ and $B$, $S^A \cup S^B$ the set of all OTUs observed and $S^A \cap S^B$ the set of OTUs observed in both samples. Then, the beta-diversity measures are defined as follows:

**Jaccard similarity index.**

$$J = \frac{|S^A \cap S^B|}{|S^A \cup S^B|} \tag{4}$$

where $|\cdot|$ denotes the number of element of a set. In particular $|S^A \cap S^B|$ is the number of species present in both sets, while $|S^A \cup S^B|$ is the total number of species (equivalent to $\gamma$-diversity). The Jaccard similarity index only accounts for presence-absence, disregarding abundance. As such, it is very sensitive to differences in rare OTUs, for which a small abundance difference could cause an OTU to go undetected in a sample but not in the other.

**Sørensen similarity index.**

$$SR = \frac{2|S^A \cap S^B|}{|S^A| + |S^B|} \tag{5}$$

where $|\cdot|$ denotes the number of element of a set. Similarly to the Jaccard similarity index, Sørensen index only accounts for presence-absence, disregarding abundance.

**Whittaker similarity index.**

$$W = \frac{2|S^A \cup S^B|}{|S^A| + |S^B|} \tag{6}$$

where $|\cdot|$ denotes the number of element of a set. The Whittaker index corresponds to the ration of $\gamma$ and $\alpha$-diversity.

**Effective Whittaker similarity.** The Effective Whittaker index is defined as the ratio of $\gamma$ and $\alpha$-diversity when they are estimated using Shannon index

$$EW = \frac{2 \exp(-\sum_i m_i \log m_i)}{\exp(-\sum_i x_i^A \log x_i^A) + \exp(-\sum_i x_i^B \log x_i^B)} \tag{7}$$

where $m_i = (x_i^A + x_i^B)/2$ and $x_i^A = n_i^A / \sum_i n_i^A$.

**Morisita-Horn similarity.**

$$MH = 2 \frac{\sum_{i \in S} x_i^A x_i^B}{\sum_{i \in S} (x_i^A)^2 + \sum_{i \in S} (x_i^B)^2} \tag{8}$$

This index includes also OTUs present in only one sample (they are counted in the denominator), but is overly sensitive to common OTUs, due to the quadratic dependence on abundance.

**Horn similarity.**

$$H = \frac{\sum_{i \in S} ((n_i^A + n_i^B) \log (n_i^A + n_i^B) - n_i^A \log (n_i^A) - n_i^B \log (n_i^B))}{(N^A + N^B) \log (N^A + N^B) - N^A \log (N^A) - N^B \log (N^B)} \tag{9}$$

where $N^A = \sum_i n_i^A$.

**Bray-Curtis dissimilarity.**

$$BC = 1 - \frac{2\sum_{i \in S} \min(n_i^A, n_i^B)}{\sum_{i \in S} n_i^A + \sum_{i \in S} n_i^B} \tag{10}$$

Similarly to the Morisita-Horn index, it includes also OTUs present in only one sample but is more sensitive to common ones.

**Dissimilarity.** The dissimilarity introduced in [10] is defined on the abundances $\hat{x}^A$ and $\hat{x}^B$, normalized on the set of OTUs common to the two samples: $\hat{x}_i = \frac{n_i}{\sum_{j \in S^{AB}} n_j}$. The dissimilarity is then computed as the root Jensen–Shannon divergence (rJSD) of $\hat{x}^A$ and $\hat{x}^B$:

$$D = \left[ \frac{D_{KL}(\hat{x}^A, m) + D_{KL}(\hat{x}^B, m)}{2} \right]^{1/2}, \qquad (11)$$

where $m = (\hat{x}^A + \hat{x}^B)/2$ and $D_{KL}(x, y) = \sum_{i \in S^{AB}} x_i \log\left(\frac{x_i}{y_i}\right)$ is the Kullback-Leibler divergence between $x$ and $y$.

This Dissimilarity measure accounts only for the differences in abundances of OTUs common to the two samples. Additionally, the measure is dominated by common OTUs, due to the $x_i$ factor in the Kullback-Leibler divergence.

**Overlap.** The overlap is the average across the two samples of the fraction of the total reads that come from OTUs observed in both samples:

$$O = \frac{1}{2} \left( \frac{\sum_{i \in S^{AB}} n_i^A}{\sum_{i \in S^A} n_i^A} + \frac{\sum_{i \in S^{AB}} n_i^B}{\sum_{i \in S^B} n_i^B} \right) = \sum_{i \in S^{AB}} \frac{x_i^A + x_i^B}{2}. \qquad (12)$$

This measure accounts both for presence-absence and for abundance. In fact, two samples have a large overlap if most of the OTUs are present in both and those that are present in only one have small abundance.

## Supporting information

**S1 Text. Supplementary sections and tables.** OTU selection and fitting a truncated log-normal distribution and estimating the total number of species. Tables with coefficient of determination $R^2$ and correlations between estimated values of $K$.
(PDF)

**S1 Fig. Relationship between Spearman correlation and carrying capacity correlation $\rho_K$ in model data.** Black points are averages of simulated data (point range corresponds to 2 standard deviations). The black curve is obtained by a quadratic fit to the data, resulting in $\rho_K \sim 0.92 + 0.34s - 0.48s^2$, where $s$ is the Spearman correlation.
(PDF)

## Author Contributions

**Conceptualization:** Silvia Zaoli, Jacopo Grilli.

**Formal analysis:** Silvia Zaoli.

**Investigation:** Silvia Zaoli, Jacopo Grilli.

**Methodology:** Silvia Zaoli, Jacopo Grilli.

**Software:** Silvia Zaoli.

**Supervision:** Jacopo Grilli.

**Visualization:** Silvia Zaoli.

**Writing – original draft:** Silvia Zaoli, Jacopo Grilli.

**Writing – review & editing:** Silvia Zaoli, Jacopo Grilli.

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
