## [Decision Letter · Decision Letter 0]

3 Jan 2022

Dear Dr Grilli,

Thank you very much for submitting your manuscript "The stochastic logistic model with correlated carrying capacities reproduces beta-diversity metrics of microbial communities" for consideration at PLOS Computational Biology.

As with all papers reviewed by the journal, your manuscript was reviewed by members of the editorial board and by several independent reviewers. In light of the reviews (below this email), we would like to invite the resubmission of a significantly-revised version that takes into account the reviewers' comments.

Both reviewers highlight the potential interest in the results, but also raise a number of questions about technical aspects and the presentation of this work. Most of these points seem valid and constructive in making the manuscript more accessible to experts in the field and a broader audience. If you decide to submit a revised version, it is important that you address all of the reviewers' concerns. As suggested by reviewer 1, it would be helpful to shorten the introduction and make the focus of this work clearer.

We cannot make any decision about publication until we have seen the revised manuscript and your response to the reviewers' comments. Your revised manuscript is also likely to be sent to reviewers for further evaluation.

Sincerely,

Tobias Bollenbach

Associate Editor

PLOS Computational Biology

James O'Dwyer

Deputy Editor

PLOS Computational Biology

Both reviewers highlight the potential interest in the results, but also raise a number of questions about technical aspects and the presentation of this work. Most of these points seem valid and constructive in making the manuscript more accessible to experts in the field and a broader audience. If you decide to submit a revised version, it is important that you address all of the reviewers' concerns. As suggested by reviewer 1, it would be helpful to shorten the introduction and make the focus of this work clearer.

Reviewer's Responses to Questions

**Comments to the Authors:**

Reviewer #1: In their manuscript “The stochastic logistic model with correlated carrying capacities reproduces beta-diversity metrics of microbial communties” the authors state that with a rather simple null-modell the change of beta diversity as a function of simple parameters of microbial communities can be predicted. The addressed question is of great relevance in (microbial) ecology. Despite the common usage of beta diversity metrics I agree with the authors that we should better understand where these values actually come from and what impacts them. The text is overall well written and the results are clearly presented. It is always great to find simple rules in complex systems because it may point towards finding something of general importance. However, sometimes you just don’t have a real scientific finding but just a general mathematical property. It remains for me a little unclear which of both is the case in the given manuscript.

Despite being familiar with theoretical modeling I am not a full theoretician myself and several of my raised points might be caused by limited understanding on my site.

Major issues:

1) My major issue with this work is that it remains a little unclear what are “real” findings at all and what are just general mathematical properties without a clear physical statement. The authors state that upon increasing the correlation between the randomly chose Ks the similarity in the obtained communities increases. It seems true for me for every function that as one plugs in more similar values also the outcomes is more similar. So I wonder what the specific modeling approach really tells me beyond this? Is the major statement that it fits the data quantitatively well? But then I feel there should be also more discussion how well the data quantitatively fits the measurements.

2) It remains a little unclear how this approach deals with potential regime shifts or more general deviations from steady states. In the introduction it is stated that the model can deal with them but it is not clear to me how. Moreover, regime shifts are removed from the data (infections). So in how far does the model assume that the system stays around a steady state? It also remains a little unclear if the K and sigma are obtained for the whole time series or just for parts of it (Fig 1C). How was the time over which K and sigma where calculated chosen? Shouldn’t the overall length of the timeseries have an impact on these values?

3) As stated by the authors sequencing data does not result in absolute abundances and therefore does not give the carrying capacities. Within a sample the measured steady state populations densities (reads) may be in the same way proportional to K but across samples this is not true. Accordingly I wonder how K from different samples can be compared (Fig1A)?

4) The authors assume that two communities from the same environment have the same sigma, but different values of K. I don’t understand this. Aren’t K and sigma estimated for all the OTUs within a community? What does K and sigma of a community then mean? Average across all OTUs? Why is K different and sigma constant across communities?

5) For the sampling two correlated values of K are drawn from a Gaussian distribution with a certain correlation. It remains for me a little unclear in how far this correlation argument is justified. Data is just shown for the correlation between two individuals in Fig 1D. That seems quite little data to make an assumption that is so central for the paper.

6) The authors state that their model explains the overlap-dissimilarity connection that can often be found but doesn’t have to be there theoretically. However, isn’t that simply a result of a detection threshold? If there is a detection threshold than for highly correlated K the two sampled species are either both above or below the threshold and thus either appear together or not with similar population densities e.g. high overlap and low dissimilarity. If K is less correlated the dissimilarity will increase because population densities become more different and it is also more likely that just one of the two falls below the detection threshold. Is there a way to better understand where this overlap-dissimilarity connection comes from?

Minor issues:

.. are present at a given locations → locations

I think it would be very useful to give line number for a manuscript otherwise it is rather difficult to refer to specific sentences.

What exactly is plotted in Figure 1? Are these binned histograms? Why is P(K) reaching around 1 at the maximum? Shouldn’t it be much smaller than that given that the integral over the plot should equal 1?

In Fig2: 100 values have the same K for both samples. Shouldn’t that result in a massive black blob at correlation=1 ?

I feel the text – especially the introduction – could be more concise and to the point. A lot of ideas seem to be introduced that seems not so central to the actual work. One has to read quite some text to find out what the goal of the paper actually. It’s of course the authors choice what they write, but I feel a little more focus would help.

In the supplementary figures are several times questionsmarks where probably should be numbers.

Reviewer #2: In the manuscript entitled “The stochastic logistic model with correlated carrying capacities reproduces beta diversity metrics of microbial communities”, Zaoli and Grilli build the null model for microbial communities using the stochastic logistic model and analyze some metrics of beta diversity. Their null model can show some patterns of beta diversity in empirical studies such as Overlap-Dissimilarity relations.

I think Result D and Figure 4 (i.e., the relationship between Overlap and Dissimilarity) are very interesting. However, I think the rests of the results need to be clarified and revised.

Major comments

1 In the current draft, it is unclear why the authors call their model the null model. I guess because the model does not include any species interactions and any other ecological properties such as migration, but please explain clearly what the authors mean by null in Introduction.

2 Result A and Figure 1

2.1 The current figures 1 a and b do not explain what the color dots and lines represent and how they are calculated. This point prevents my understanding in section Result A.

2.2 In the main text, the authors argue that “as shown in Fig.1 C, the estimation of K … are strongly correlated” (the first paragraph of Result A) and that “however the correlation is lower than the temporal one (Fig 1D)” (the second paragraph of Result B), they do not show any correlation coefficients. I think that they should show for example Pearson or Spearman correlations.

3 Result B: I have some concerns in this section.

3.1 First, the section title seems misleading. In this section, the authors show the scatter plots of Pearson correlations of species abundances between two environments and other metrics of beta diversity obtained from their null model. The authors do not predict anything here.

3.2 Second, the choice of the metrics of beta diversity seems problematic. Although some measures (e.g., Bray-Cruits dissimilarity and Jaccard similarity) are widely used in (microbial) ecology, I have never seen the studies using Pearson correlation as beta diversity (see also the next comment). In addition, there are many metrics of beta diversity that the authors do not analyze but are widely used (e.g., Sørensen index, the ratio of gamma diversity to alpha diversity, or gamma diversity minus alpha diversity). See, for example, Beck et al (2013) https://onlinelibrary.wiley.com/doi/10.1111/2041-210x.12023, Tuomisto (2010), https://onlinelibrary.wiley.com/doi/10.1111/j.1600-0587.2009.06148.x, or Barwell et al (2015), which the authors cite (ref. 24). The authors need to reconsider the choices of the metrics and/or to justify them (e.g., choosing metrics that are widely used in recent studies).

3.3 Related to the above two points, I am not sure whether the x-axes of figure 2 (and figure 3) should be Pearson correlation of species abundances. If the authors want to predict something, I think they should plot the (each measure of) beta diversity in the empirical data against the beta diversity predicted by the null model tuning parameter rho_K. As I do not think Pearson correlation is used as beta diversity, figure 2 (and figure 3) does not make sense to me.

4 In addition to the above comment, I have a concern about figure 3 and the sentence “The relationship predicted by the null model follows in a remarkably precise manner the empirical patterns” in the first paragraph of section Result C. Of course, the pattern of the mean values in the null model (red dots in figure 3) seems similar to the patterns of empirical data. However, if we compare the exact points from the null model (i.e., figure 2) with the empirical ones (figure 3), I am not sure whether the distributions of dots in the null model and empirical data are similar. It seems to me that the distributions of the dots at least in panels B, C, and D differ between figures 2 and 3. The authors need to quantify the similarities between the null model and the empirical data.

Minor comments

1. Figure legend of 1a. I recommend that the authors add the explanation on the dashed line. Although it is explained in the main text, the figure legend should be (ideally) clear without reading the main text. In addition, the figure legend should explain what the colored dots and lines represent in panels A and B.

2. I think there is a typo in r.h.s. of equation (2): 1/(tau)*lambda*(1- lambda/K)+…

3. In Method D and Result B, adding the indexes to the beta diversity measurements could make the first sentence clearer. I mean (i)Pearson correlation, (ii) Jaccard similarity index, (iii) Morisita-Horn dissimilarity, (iv)Brays-Cruits dissimilarity…. In addition, Result B says the authors analyze seven metrics while Method D says they analyze six measures. This is inconsistent.

**Have the authors made all data and (if applicable) computational code underlying the findings in their manuscript fully available?**

Reviewer #1: Yes

Reviewer #2: Yes

PLOS authors have the option to publish the peer review history of their article (what does this mean?). If published, this will include your full peer review and any attached files.

Reviewer #1: No

Reviewer #2: No
---

## [Decision Letter · Decision Letter 1]

21 Mar 2022

Dear Dr Grilli,

We are pleased to inform you that your manuscript 'The stochastic logistic model with correlated carrying capacities reproduces beta-diversity metrics of microbial communities' has been provisionally accepted for publication in PLOS Computational Biology.

Best regards,

Tobias Bollenbach

Associate Editor

PLOS Computational Biology

James O'Dwyer

Deputy Editor

PLOS Computational Biology

Both reviewers appreciate the improvements made in the revision and support publication. Please consider all remaining suggestions they made when preparing the final version of the manuscript.

Reviewer's Responses to Questions

**Comments to the Authors:**

Reviewer #1: The authors clarified my concerns and from my side the work can be published. There are a few smaller comments which the the authors should take as suggestions.

It took me some time to understand Fig.1 . I think It would help to write a little clearer what the figure shows and especially what the different colors, lines and dots mean. I assume that is stated in the SciAdv publications, but maybe would be nice to have the information here as well?

Fig 4: sampled in both → maybe add word communities? It was a little unclear at first what sampled in both/just one/none means.

line 158: as → has

Congratulations for this work!

Reviewer #2: I really appreciate the effort the authors made to revise this manuscript. My previous comments were mainly about Results B-C and Figures 2 and 3. These are now clearer than before and I am satisfied with the responses from the authors. I have one very minor comment.

Line 217-218: the authors write "In SI Table S1, we report the statistical significance of the deviation of the data respect to the model", but this sentence seems strange to me. I do not think that the authors should mention the statistical significance here. Many researchers use statistical significance based on p-values calculated by certain statistical tests (e.g., t-test). On the other hand, Table S1 shows R^2, a goodness of fit of a linear model, and large R^2 does not necessarily mean small p-values for coefficients of the focal linear model. I do not think that the sentence from line 217 to 218 is needed: reporting R^2 is sufficient to argue that the model predicts the metrics of beta diversity in natural communities.

**Have the authors made all data and (if applicable) computational code underlying the findings in their manuscript fully available?**

Reviewer #2: Yes

PLOS authors have the option to publish the peer review history of their article (what does this mean?). If published, this will include your full peer review and any attached files.

Reviewer #2: No

---

## [Editor Report · Acceptance letter]

28 Mar 2022

PCOMPBIOL-D-21-02238R1 

The stochastic logistic model with correlated carrying capacities reproduces beta-diversity metrics of microbial communities

Dear Dr Grilli,

I am pleased to inform you that your manuscript has been formally accepted for publication in PLOS Computational Biology. Your manuscript is now with our production department and you will be notified of the publication date in due course.

With kind regards,

Zsofia Freund
